# Low prevalence of current and past SARS-CoV-2 infections among visitors and staff members of homelessness services in Amsterdam at the end of the second wave of infections in the Netherlands

Ellen Generaal[1,2,3,4☯]*, D. K. (Daniela) van Santen[1,3,4,5,6☯], Sophie L. Campman[1,3,4], Marjolein J. Booij[1], Dylan Price[7], Marcel Buster[7], Christa van Dijk[7], Anders Boyd[1,3,4,8], Sylvia M. Bruisten[3,4,7], Alje P. van Dam[3,4,7], Mariken van der Lubben[7], Yvonne T. H. P. van Duijnhoven[1,9], Maria Prins[1,3,4]

1 Department of Infectious Diseases, Research and Prevention, Public Health Service of Amsterdam, Amsterdam, the Netherlands, 2 National Institute for Public Health and the Environment (RIVM), Centre for Infectious Disease Control (Cib), Bilthoven, the Netherlands, 3 Amsterdam UMC Location University of Amsterdam, Infectious Diseases, Amsterdam, the Netherlands, 4 Amsterdam Institute for Infection and Immunity, Infectious Diseases, Amsterdam, the Netherlands, 5 Department of Disease Elimination, Burnet Institute, Melbourne, Australia, 6 Department of Epidemiology and Preventive Medicine, Monash University, Melbourne, Australia, 7 Public Health Service of Amsterdam, Amsterdam, the Netherlands, 8 Stichting HIV Monitoring, Amsterdam, the Netherlands, 9 Public Health Service of Rotterdam, Rotterdam, the Netherlands

☯ These authors contributed equally to this work.

* egeneraal@ggd.amsterdam.nl

**Data Availability Statement:** The datasets generated and/or analysed during the current study

# Abstract

## Background

People experiencing homelessness (PEH) may be at increased risk of SARS-CoV-2 infection and severe COVID-19. The Dutch government established emergency shelters and introduced preventive measures for homelessness services. There were no major SARS-CoV-2 outbreaks noticed among PEH during the first two waves of infections. This study aimed to assess the prevalence of current and past infections among PEH and staff by conducting an on-site COVID-19 screening project at homelessness services in Amsterdam, the Netherlands.

## Methods

We assessed the proportion of visitors and staff members of four homelessness services at two locations in Amsterdam with positive SARS-CoV-2 qPCR and antibody results (IgG/IgM Rapid Test/Biozek) in May 2021. We also assessed sociodemographic, clinical and lifestyle characteristics, compliance with basic prevention measures and intention to vaccinate against COVID-19 among PEH and staff.

are not publicly available due to national privacy policies but are available from the corresponding author on reasonable request. For data related requests, please contact Dr Ellen Generaal, Email: egeneraal@ggd.amsterdam.nl. For long term request access, please contact the data management team (Email contact: datamanagersoz@ggd.amsterdam.nl). This team will have permanent access to the dataset used for our study.

**Funding:** The authors received no specific funding for this work.

**Competing interests:** The authors have declared that no competing interests exist.

**Abbreviations:** *CI*, Confidence interval; *IgG*, Immunoglobuline G; *IgM*, Immunoglobine M; *IQR*, Interquartile range; *N.A.*, Not asked; *PCR*, Polymerase chain reaction; *PEH*, People experiencing homelessness; *qPCR*, Quantitative polymerase chain reaction; *SARS-CoV-2*, Severe Acute Respiratory Syndrome Coronavirus; *UK*, United Kingdom.

## Results

A total of 138 visitors and 53 staff members filled out a questionnaire and were tested. Among PEH, the SARS-CoV-2 positivity rate was 0% (0/133;95%CI = 0–1.9) and the antibody positivity rate was 1.6% (2/131;95%CI = 0.8–7.5) among those without prior COVID-19 vaccination. Among staff, these percentages were 3% (1/32;95%CI = 0.1–16.2) and 11% (5/53;95%CI = 3.6–23.6), respectively. Most participants were often compliant with the basic preventive measures 'not shaking hands', 'wearing a face mask' and 'washing hands', but not with 'physical distancing'. High vaccination intent was more common among staff members (55%) than among visitors (42%), while high trust in the governmental COVID-19 policies was more common among visitors (41%) than among staff (30%).

## Conclusions

We observed a low prevalence of past and current SARS-CoV-2 infections among PEH, which may be explained by instated shelter policies, limited daily activities of PEH and compliance with prevention measures. Vaccine hesitancy and mistrust among visitors and staff could hinder vaccination uptake, suggesting that interventions towards homelessness services are needed.

## Background

People experiencing homelessness (PEH) may be at increased risk of SARS-CoV-2 infection and severe COVID-19 disease due to certain attributes of their lifestyle, which include lack of living accommodation, increased prevalence of chronic and physical conditions, and limited access to health care [1]. Moreover, the lockdown measures used to control SARS-CoV-2 infections resulted in diminished access to services for PEH in many countries [2].

Studies conducted during the first epidemic wave of SARS-CoV-2 infections in the US showed higher PCR positivity rates (i.e. current infection) among PEH than among the general population, with 16% testing positive in San Francisco [3] and 36% in Boston [4]. These studies were however carried out in centers where COVID-19 outbreaks were taking place. A study in Marseille, France, conducted during the same time period indicated a somewhat lower SARS-CoV-2 PCR positivity rate in a subsample of PEH, at 9% [5]. The highest SARS-CoV-2 PCR positivity rate was found among PEH in homelessness facilities where overcrowding from sharing of dormitories and communal (e.g. shower) facilities was common [5, 6]. A study conducted across nine Dutch cities during the first wave reported a SARS-CoV-2 PCR positivity rate of 17% among PEH with COVID-related symptoms or who had a close contact with individuals with COVID-19 and visited the 'street doctors' [7], which was higher than the rate found among the general population during that period [8]. In contrast, a population-based study from Wales, United Kingdom (UK) using routinely collected administrative data found that the SARS-CoV-2 PCR positivity rate among PEH was comparable to that of the general population (5.0% versus 5.6% between March 2020 and March 2021) [9]. The authors suggest that this low percentage may have been due to the policy pandemic response for PEH, moving away from communal accommodation to offering PEH private rooms to reduce the risk of SARS-CoV-2 transmission [9].

In the Netherlands, no major outbreaks among visitors attending homelessness services had been identified since the start of the pandemic until December 2021, contrary to the situation in the US. It might be that the extra strategies set in place to control COVID-19 among

PEH during the lockdown periods, such as those implemented in the UK [9], resulted in limited transmission. Dutch municipalities provided emergency accommodations with more space for all PEH, irrespective of insurance and legal status, to be able to accommodate people who were unable to follow the "stay-home" measure [10]. In addition, isolation and quarantine accommodations were offered to PEH who were infected with SARS-CoV-2 and to close contacts of infected individuals in homelessness services [10]. However, infections in homelessness services may have stayed under the radar because PEH may be less likely to receive testing [11]. As PCR testing of PEH may have occurred less frequently than among the general population, it is relevant to evaluate the prevalence of past SARS-CoV-2 infections in this group and among staff members working in the homelessness services who are frequently in close contact with PEH, in particular because COVID-19 continues to spread around the world and new SARS-CoV-2 variants of concern may evolve. Given that PEH may be at increased risk for severe COVID-19 due to chronic physical and mental conditions impaired by substance or alcohol abuse [7, 12, 13] and that SARS-CoV-2 transmission may occur rapidly in places with overcrowding [5, 6], studying the prevalence of past SARS-CoV-2 infections in homelessness services during the COVID-19 pandemic is highly relevant.

Therefore, we conducted an on-site screening project at homelessness services in Amsterdam during May 2021, when the SARS-CoV-2 delta variant was most dominant in the Netherlands. We aimed to assess the proportion of visitors and staff of homelessness services with positive SARS-CoV-2 PCR and antibody results, and sociodemographic, clinical and lifestyle determinants of current and past infection. In addition, we measured daily activities, compliance with preventive measures and intention to vaccinate against COVID-19 among visitors and staff members.

## Methods

### Study design

We conducted a cross-sectional study among PEH, either living on the streets or residing in temporary shelters, and staff members working in homelessness services in Amsterdam, the Netherlands. We invited PEH and staff attending two locations for PEH in Amsterdam to participate in our study. One location included one homelessness service: a walk-in center (approximately 50 visitors per day). The other location included three homelessness services: a walk-in center (approximately 50 visitors per day), temporary housing (approximately 54 visitors per day) and emergency accommodation during lockdowns (approximately 54 visitors per day).

### Eligibility criteria

Inclusion criteria were being a visitor/resident or staff member from the homelessness service, being 18 years or older, being proficient in Dutch, English or Polish and being able to understand the study information. Exclusion criteria were unable to hold an interview and/or provide informed consent or having severe COVID-19 related symptoms (including fever and shortness of breath), as perceived by the staff. At one testing location, including the majority of participants, staff members pre-selected visitors who had inclusion criteria. For this reason, and because we had an onsite translator, we did not have to exclude any respondent at the start of our study.

### Ethics approval and consent to participate

All participants provided written informed consent. Research information and consent forms were translated into English and Polish, and a Polish-to-English translator was present during

the study. The project was conducted according to the ethical guidelines of the 1975 Declaration of Helsinki. The local medical ethics committee of Amsterdam UMC, location AMC approved the research project (NL76623.018.21).

## Study procedures

Pre-study announcements (posters) were distributed a few weeks before the study. Between May 3 and May 21, 2021, we actively recruited visitors and staff members of the homelessness services, with the assistance of the coordinators of the centers. Visitors participated voluntarily and participants received a small gift plus 5 euros as incentive. Since major groups of PEH in Amsterdam are from Polish origin, a Polish translator was present during the study. Participants were able to opt-out for PCR testing, antibody testing or the questionnaire.

Participants underwent a single throat plus nasopharyngeal swab for qPCR-testing inside a 'testing bus' (S1 Fig). The qPCR targeted the E gene and the N genes with an internal control for correct swab sampling. It was validated at the Public health laboratory of GGD Amsterdam and performed using a modified protocol with the RotorGene platform (Qiagen, Hilden, Germany), as described previously [14]. In addition, two drops of finger blood were collected via finger-stick for antibody testing using the COVID-19 IgG/IgM Rapid Test Cassette in a tent next to the bus (Biozek Medical, Apeldoorn, the Netherlands).

After obtaining specimens for testing, individuals completed a short questionnaire with information about socio-demographics, lifestyle, (changes in) daily activities since the pandemic, COVID-19 testing, disease history, COVID-19 vaccination status, and comorbidities including lung problems, chronic heart disease, diabetes, severe kidney or liver disease, impaired resistance, infections with HIV, hepatitis B or hepatitis C or obesity.

Compliance with COVID-19 prevention measures, including not shaking hands, wearing a face mask, washing hands and keeping 1.5 meters distance were measured on a 7 point Likert-scale, with responses ranging from 1 ('never') to 7 ('always'). Vaccination intention was defined by two statements: (1) 'Are you planning to get vaccinated against the coronavirus when it is your turn to be vaccinated?' and (2) 'What is the likelihood that you will actually be vaccinated against the coronavirus?'. Responses ranged from 1 ('no, absolutely not' and 'very unlikely') to 7 ('yes, absolutely' and 'very likely') on a 7-point Likert scale. We combined the responses of the two statements as a mean score to create a more accurate measure of vaccination intent, based on Ajzen's Theory of Planned Behavior which states that both behavioral intentions and perceived behavioral control can predict behavior [15]. Additionally, the internal consistency of the two statements demonstrated high reliability (Cronbach's $\alpha$ = 0.95). Vaccination intention was categorized into low (mean score 1.0–2.5), medium (3.0–5.0) and high (5.5–7.0), based on the distribution of the combined score (S2 Fig) and in line with another study [16]. Trust in the governmental policy response to the pandemic was defined by the statement 'How much do you trust the way the Dutch government is trying to keep the coronavirus under control', with responses ranging from 1 ('absolutely no trust') to 5 ('a great deal of trust') on a 5-point Likert scale. This variable was categorized into low (score 1 or 2), neutral (score 3) and high (score 4 or 5).

## Statistical analyses

Descriptive baseline characteristics were reported as means, median or percentages. We calculated the proportion of persons with a positive PCR-test result and the proportion with a positive antibody test result among those with a valid test result. 95% confidence intervals (CI)s were calculated using either the Clopper-Pearson or Jeffrey's method, depending on

whether the proportion was ≥0.02 or <0.02, respectively. Since a positive SARS-CoV-2 antibody test result may refer to either past infection or vaccination, we conducted sensitivity analyses for (i) visitors and (ii) staff members with a valid antibody result, presenting antibody results with exclusion of vaccinated persons and persons with an unknown vaccination status.

## Results

### Description of the study sample

We included a total of 138 visitors and 53 staff members of the two locations with homelessness services for PEH in Amsterdam. Table 1 shows the demographics of our study population stratified by participant type. Visitors and staff members had a median age of 44 (IQR = 37–51) and 40 (IQR = 35–52) years, respectively. The majority of visitors and staff members were men (94% and 67%, respectively). Most visitors were from Eastern/Central European origin (46%), and most staff members (62%) were from Dutch origin. Additional information on sleep locations and lifestyle characteristics of the visitors can be found in S1 Table. The majority of visitors resided in a temporary homeless shelter (36.3%), emergency accommodation (28.9%) or the streets (17.8%). Regular use of cannabis was reported by 40.9%, and regular use of hard drugs was reported by 11.4% of visitors. Daily or weekly use of alcohol was reported by 46.3% of visitors.

**Table 1. Sociodemographic and lifestyle characteristics of visitors and staff members of homeless shelters in Amsterdam, the Netherlands, May 2021.**

|  | Visitors (n = 138) | Staff members (n = 53) | Total sample (n = 191) [2] |
|---|---|---|---|
|  | n (%) | n (%) | n (%) |
| *Sociodemographic and lifestyle characteristics* |  |  |  |
| **Age in years**, median [IQR] | 44.0 [37.0–51.0] | 40.0 [35.0–52.0] | 43.0 [36.0–51.0] |
| **Sex** |  |  |  |
| Female | 8 (6.0) | 17 (32.7) | 25 (13.4) |
| Male | 126 (94.0) | 35 (67.3) | 161 (86.6) |
| **Region of birth** |  |  |  |
| The Netherlands | 11 (8.2) | 33 (62.3) | 44 (23.5) |
| Africa | 29 (21.6) | 7 (13.2) | 36 (19.3) |
| Eastern/Central Europe | 62 (46.3) | 6 (11.3) | 68 (36.4) |
| Southwest Asia | 9 (6.7) | 2 (3.8) | 11 (5.9) |
| South America, Caribbean/Antilles | 15 (11.2) | 3 (5.7) | 18 (9.6) |
| Other [1] | 8 (6.0) | 2 (3.8) | 10 (5.3) |
| **Gone to work in past 7 days** |  |  |  |
| Never | 106 (78.5) | 3 (5.7) | 109 (58.0) |
| Sometimes (1–6 times) | 27 (20.0) | 45 (84.9) | 72 (38.3) |
| Daily (7 times) | 2 (1.5) | 5 (9.4) | 7 (3.7) |
| **Injecting drug use** |  |  |  |
| No | 108 (81.8) | N.A. | N.A. |
| Yes, former | 17 (12.9) | N.A. | N.A. |
| Yes, recent (past month) | 7 (5.3) | N.A. | N.A. |

Abbreviations: *IQR* interquartile range; N.A. = not asked.

[1] Other = Western Europe (not the Netherlands), USA and Australia

[2] N-values may vary due to missing variables (visitors: n = 4 for age, n = 4 for sex, n = 4 for region of birth, n = 3 for gone to work, n = 6 for injecting drug use; staff: n = 1 for sex)

## Prevalence of SARS-CoV-2 infection and antibodies

Among visitors, the prevalence of SARS-CoV-2 infection was 0% (0/133, 95%CI = 0–1.9) and the prevalence of SARS-CoV-2 antibody positivity was 3% (4/134, 95%CI = 0.8–7.5) for the total sample and 1.6% (2/131, 95% = 0.3–5.0) when excluding vaccinated visitors (Table 2). Among staff members, the prevalence of SARS-CoV-2 infection was 3% (1/32, 95%CI = 0.1–16.2) and the prevalence of SARS-CoV-2 antibody positivity was 11.3% (6/53, 95%CI = 4.3–23) and 10.9% (5/46, 95%CI = 3.6–23.6) when excluding vaccinated staff (Table 2). Analysis examining the determinants of current and past SARS-CoV-2 infections was precluded by the small number of infections in our study population.

## Clinical symptoms related to COVID-19

A large proportion of visitors and staff (44.1% and 41.5% of the total sample, respectively) reported at least one COVID-related clinical symptom during the survey, such as a nasal cold or a cough, irrespective of infection (Fig 1). The one SARS-CoV-2 PCR-positive individual reported no respiratory or other COVID-related symptoms. Five out of seven (71.4%) SARS-CoV-2 antibody positive individuals reported at least one clinical symptom, and this percentage was similar among antibody negative individuals (67.6% reported at least one symptom). Twenty eight percent of visitors and 9% of staff members reported at least one comorbid condition, mainly chronic lung problems.

**Table 2. SARS-CoV-2 PCR and antibody test results and vaccination status among visitors and staff members of homeless shelters in Amsterdam, the Netherlands, May 2021.**

| | Visitors (n = 138) | Staff members (n = 53) | Total sample (n = 191) |
|---|---|---|---|
| | n (%, 95%CI) | n (%, 95%CI) | n (%, 95%CI) |
| *SARS-CoV-2 PCR* | | | |
| Positive test | 0 (0, 0–1.9) | 1 (3.1, 0.1–16.2) | 1 (0.6, 0.1–2.8) |
| Declined testing [1] | 5 | 21 | 26 |
| *SARS-CoV-2 antibodies* | | | |
| Positive test | 4 (3.0, 0.8–7.5) | 6 (11.3, 4.3–23.0) | 10 (5.3, 2.6–9.6) |
| Positive test, without being vaccinated[2] | 2 (1.6, 0.3–5.0) | 5 (10.9, 3.6–23.6) | 7 (4.0, 1.6–8.2) |
| Declined testing/invalid result [1] | 4 | 0 | 4 |
| **COVID-19 vaccination status** [3] | | | |
| No | 128 (94.8) | 41 (85.4) | 169 (92.3) |
| Partly | 1 (0.7) | 4 (8.3) | 5 (2.7) |
| Fully | 2 (1.5) | 3 (6.3) | 5 (2.7) |
| Partially but unknown vaccine | 4 (3.0) | 0 (0) | 4 (2.2) |
| Unknown | 3 | 5 | 8 |

Abbreviations: *SARS-CoV-2* Severe Acute Respiratory Syndrome Coronavirus; *PCR* Polymerase chain reaction; *CI* confidence interval.

[1] From the total study sample, 5 visitors and 21 staff members declined participation in PCR testing, resulting in a total of n = 133 for visitors and n = 32 for staff. A total of 3 visitors declined participation in antibody testing, and one visitor had an invalid test result, resulting in a total of n = 134 for visitors and n = 53 for staff.

[2] Fourteen participants were vaccinated against SARS-CoV-2 (3 visitors and 7 staff members) or had an unknown vaccination status (4 visitors) and were therefore excluded in this analyses.

[3] The variable vaccination status has 8 missings (3 for clients, 5 for staff). Participants were partially vaccinated when they received 1 dose of Pfizer, Moderna or AstraZeneca, and fully vaccinated when they received 2 doses of Pfizer, Moderna or AstraZeneca, or 1 dose of Janssen. The vaccination status 'partly but unknown vaccine' refers to if participants had one dose of an unknown vaccine.

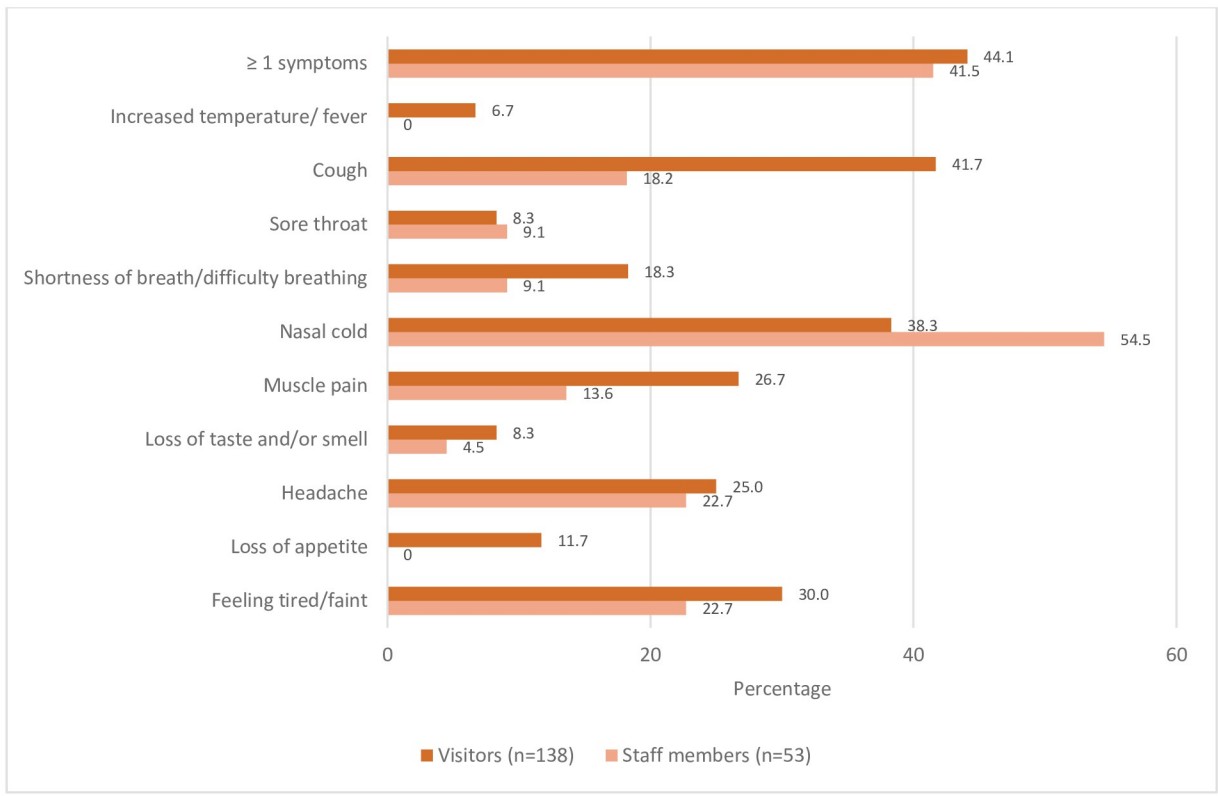

**Fig 1. Self-reported COVID-19 related symptoms\* among visitors and staff members of homeless services over the past 7 days in Amsterdam, the Netherlands, May 2021.** \* Missing variables for symptoms: n = 2 for visitors, 0 for staff.

## Potential exposure to COVID-19: Daily activities and previous testing

Overall, visitors seem to report fewer daytime activities and contact patterns, such as going to work or visiting friends, than staff (S2 Table). Thirty three percent (33%) of visitors reported to have lost their paid work and 37% of visitors spent more time at the shelter than before the COVID-19 pandemic (S3 Table). Among visitors, 41% (55/134) reported previous testing for SARS-CoV-2 infection and 5.9% (8/135) a previous infection not confirmed with a test (self-report). Of those, 1 visitor also tested antibody positive in our study. Among staff members, 30% (16/53) reported previous testing, 40% (21/53) a previous infection not confirmed with a test, and 15% (8/53) a previous infection confirmed with a test (self-report). Of those, 4 staff members tested antibody positive in our study.

## Compliance with preventive measures

The majority of visitors and staff were frequently or regularly compliant with the basic COVID preventive measures 'not shaking hands', 'wearing a face mask' and 'washing hands' (Fig 2A and 2B). Not all visitors and staff were compliant with the measure of 'keeping 1.5 meters distance': 23.9% and 13.2%, respectively, reported never or seldom keeping 1.5 meters distance to other people (i.e., frequent close contacts) in the past 7 days.

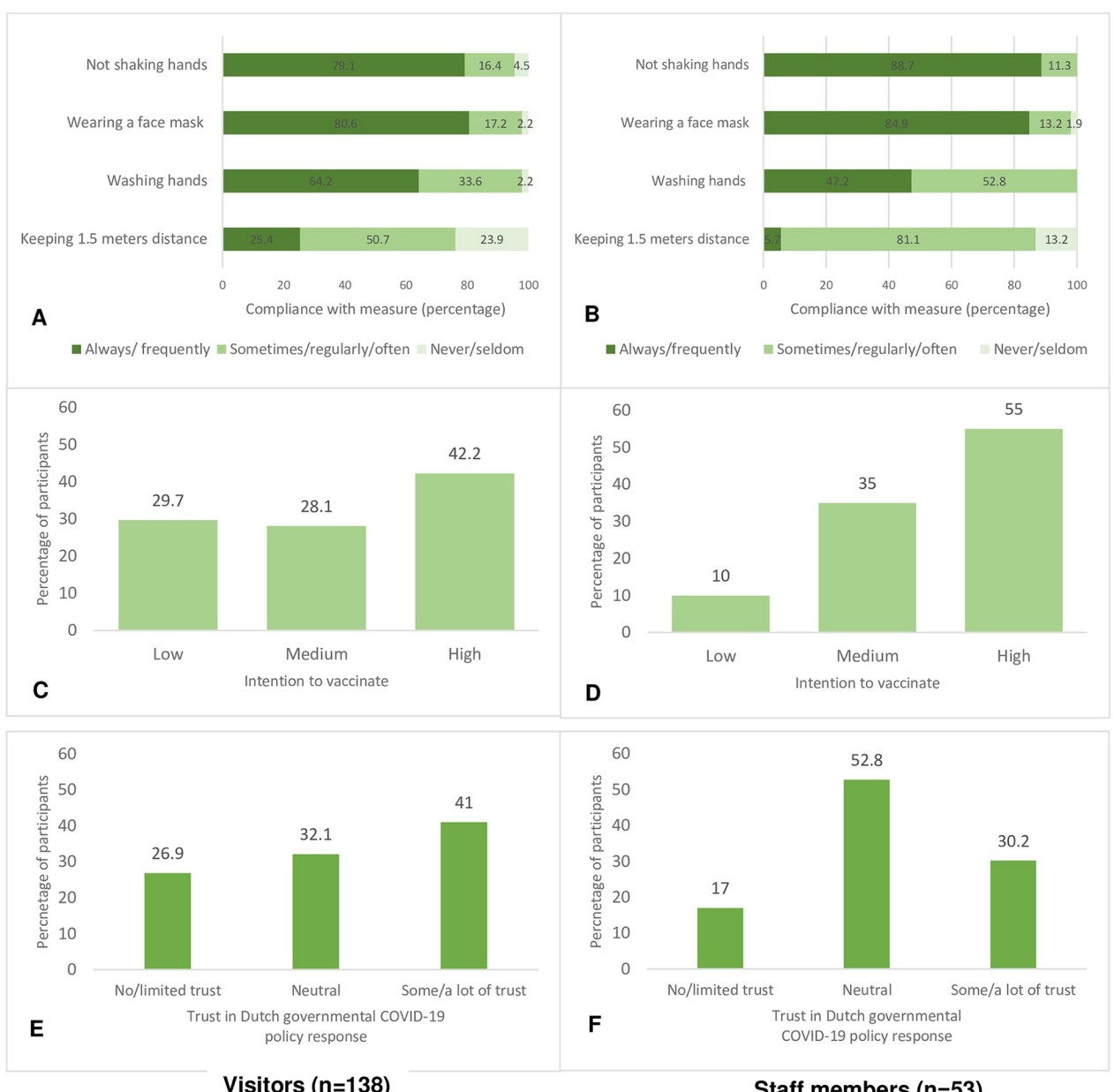

**Fig 2. Compliance with prevention measures, intention to vaccinate and trust in COVID-19 policy response among visitors and staff members of homeless services.** A) Compliance with the basic COVID-19 prevention measures (in the past 7 days) for **visitors** (n = 4 missing variables). (B) Compliance with the basic COVID-19 prevention measures (in the past 7 days) for **staff**. (C) Intention to vaccinate against COVID-19 for **visitors** (n = 4 missing variables). (D) Intention to vaccinate against COVID-19 for **staff** (n = 1 missing variable). (E) Trust in the Dutch governmental COVID-19 policy response for **visitors**. (F) Trust in the Dutch governmental COVID-19 policy response for **staff**.

## Intention to vaccinate and (mis)trust in governmental COVID-19 policy

Intention to vaccinate against COVID-19 was lower among visitors than among staff members: 57.8% of visitors and 45.0% of staff scored 'low' or 'medium' (Fig 2C and 2D), the remaining participants scored 'high' (42.2 of visitors and 55% of staff). In addition, trust in the governmental policy response to the COVID-19 pandemic was reported as 'no or limited

trust' for 27% of visitors and for 17% of staff members (Fig 2E and 2F). 41.0% of visitors and 30.2% of staff members had 'some or a lot of trust' in the governmental policy response, the remaining participants (32.1% of visitors and 52.8% of staff) scored 'neutral' (Fig 2E and 2F).

## Discussion

This cross-sectional study reports on the proportion of visitors and staff of homelessness services who were SARS-CoV-2 PCR and antibody positive in Amsterdam during May 2021, at the end of the second wave of infection in the Netherlands. The proportion of participants with current and past SARS-CoV-2 infection was low (0% and 1.6%, respectively) among visitors, and somewhat higher among staff members (3% and 10.9%). These findings are in contrast to earlier studies from the Netherlands [7], France [5] and the US [3, 4] conducted during the first wave, showing high PCR positivity rates for SARS-CoV-2 infection among PEH. Notably, these studies either included PEH suspected of COVID-19 [7] or recruited PEH at shelters where a COVID-19 cluster was identified [3, 4] or during the peak of infections [5], all of which likely resulted in higher PCR positivity estimates than during a non-peak period. This was shown in later studies at shelters in the US where no or only one previous PCR positive case was identified [17], resulting in a substantially lower SARS-CoV-2 PCR positivity rate among PEH: 5% in Seattle, Washington [18]; 4% in Atlanta, Georgia [17], 2% in King-County, Washington [18].

Our results are in line with a population level study from Wales, UK showing lower PCR positivity among PEH as compared to the general population [9]. The authors explain this finding by a proactive policy response directing local authorities on sourcing additional temporary accommodation for PEH, including, for instance, hotels and bed and breakfasts accommodation and adapted night-shelters, a situation comparable to the Netherlands. Amsterdam had less densely homelessness services during the pandemic (e.g., limited numbers of PEH) and different shelter policies (e.g., sufficient resources to prevent overcrowding) than other cities outside of the Netherlands. Earlier research in the UK suggests that single-room accommodation and heightened infection prevention methods are important strategies to prevent COVID-19 disease in homeless populations [19]. Moreover, since the Dutch community of PEH is relatively small, estimated at around 32,000 in 2021 [20] and 5000 in Amsterdam [21], access to homeless communities is relatively easy in the Netherlands, contrary to the inability to reach PEH as reported in other countries such as the US [22]. Amsterdam has a wide range of homelessness services available on a walk-in basis (also during the day) for all PEH independent of legal status, including both EU and non-EU citizens, to prevent rough sleeping and improve access to health care for PEH. There is also an ongoing collaboration of homelessness services with the local Public Health Services, which may have contributed to implementing preventive strategies in the shelters.

Another explanation for the low percentage of participants with current SARS-CoV-2 infections is the drop in SARS-CoV-2 incidence in Amsterdam during our study period, which took place at the end of the second lockdown and which had significant impact on reducing individual activities and travel behavior [23]. Other explanations for the low SARS-CoV-2 (sero)prevalence among PEH could be the limited contact patterns of PEH, relativity good compliance with preventive measures, the increased spacing between beds and reduced client-to-client contact during the lockdowns and decreased daytime activities at homelessness services.

Our overall antibody seroprevalence (including asymptotic past infection) estimate was 1.6% among PEH and 11% among staff (who were not vaccinated), as compared to 12% in the general population in the Netherlands [24]. In a prospective cohort study in Amsterdam, the

cumulative incidence of SARS-COV-2 among residents ranged from 15.9% for Dutch-born individuals to 64.6% for Ghanaian-born individuals at the end of March 2021 [25]. The proportion of past infection (i.e. SARS-CoV-2 2 antibodies) in our study was also substantially lower than found in other countries such as France [26, 27]. A cross-sectional study conducted during the lockdown of the first wave in Paris, SARS-CoV-2 seroprevalence ranged from 28% to 89% among 818 PEH and was highest among PEH who lived in densely populated worker residences [26]. Another community-based study in June 2020 among 1,156 PEH in Marseille, France showed antibody positivity ranging from 2.2% in people who lived on the streets to 8.1% in people living in emergency shelters [27], which is more in line with our findings. Considering the mixed results from these studies, SARS-CoV-2 (sero-)prevalence needs to be evaluated with at a country-specific level and monitored at different timepoints among PEH.

The low proportion of past SARS-CoV-2 infections for PEH is notable, although this could also be explained by the same reasons as described regarding results from the PCR prevalence, that is, less exposure to the virus due to a solitary life and the situation in the homelessness services in Amsterdam, i.e. the governmental and personal preventive measures. The proportion of past SARS-CoV-2 infections based on self-report was substantially higher than based on antibody testing. This difference could be due to recall or reporting bias of participants. On the other hand, we may have underestimated the proportion of participants with a past infection since later studies showed that the point-of-care (POC) test that we used in our study had relatively low sensitivity for individuals with mild or asymptomatic infections, estimated at 85.7% (95CI = 65.4–95.0) [28]. In addition, the waning of antibodies post-infection occurs over time and POC tests have shorter detectability windows than other serologic laboratory assays [29], limiting comparison of our result for antibodies with the Dutch general population. Future studies should incorporate more accurate testing than point-of-care testing to study the proportion of past SARS-CoV-2 infections in a community setting.

Our study further showed that COVID-19 related symptoms were frequently reported among both PEH and staff members in our study, irrespective of being infected with SARS-CoV-2. This is in line with earlier findings that COVID-19 related symptoms among PEH are a weak predictor of current SARS-CoV-2 infection [4] and that many PEH can have SARS-CoV-2 infections without reporting symptoms [5]. Hence, screening aymptomatic individuals during periods with peaks in COVID-19 cases could be considered, particularly in places where overcrowding cannot be avoided.

The low or medium vaccination intention and the mistrust in the government's SARS-CoV-2 policy response warrants further attention, particularly because this study took place before the Dutch government offered vaccination to visitors of homelessness services. This may have led to a lower vaccination uptake among PEH but no data are currently available to evaluate uptake in this group. Hence, examining whether this population is sufficiently protected for COVID-19 severe disease and mortality by infection is warranted, particularly as new variants of concern continue to circulate.

Our study has several limitations. First, testing represents a single time point at the end of the second wave of infections including a relatively small sample of PEH and staff members attending two sites for PEH in Amsterdam. A study among a larger group of PEH during a different time period of restrictions may result in different estimates. Second, the acceptance rate of our study is unclear because participants were recruited with the assistance of coordinators of the centers who did not register the number of individuals invited to participate and the reasons for non-participation. Third, whether our results are generalizable to other PEH in the Netherlands cannot be guaranteed, however we did make an effort to reduce selection bias by including both walk-in centers (more frequently visited by people living on the streets) and temporary housing for PEH, aiming to include a diverse group of PEH in our study.

In conclusion, this was the first study estimating current and past SARS-CoV-2 infections among PEH and staff of homelessness services in the Netherlands. A major facilitating factor for this screening study was the good accessibility of homeless communities in the Netherlands. Our study suggests that the additional preventive policy measures in homelessness services in Amsterdam may have resulted in limited COVID transmission among PEH, although the effect of having less contacts and a more solitary life than the general population could not be ruled out. Therefore, as COVID-19 continues to circulate around the world, emergency services for PEH should be made rapidly accessible when needed, and vaccination uptake should be evaluated, in particular because PEH are regarded to be at increased risk of severe COVID-19 due to comorbidities [7, 12], including chronic infections such as hepatitis C [30], and because we found vaccination intention among PEH to be relatively low. There is growing evidence that, in addition to general preventive measures, public policies should offer adequate shelter to PEH to prevent outbreaks of infectious diseases [31].

## Supporting information

**S1 Table. Additional characteristics related to homelessness and lifestyle of visitors of homeless services in Amsterdam, the Netherlands.**
(DOCX)

**S2 Table. Daytime activities (potential COVID exposure) of visitors and staff members of homeless services in Amsterdam, the Netherlands, May 2021.**
(DOCX)

**S3 Table. SARS-CoV-2 previous testing, impact of the pandemic on daily activities for visitors and staff members of homeless services in Amsterdam, the Netherlands, May 2021.**
(DOCX)

**S1 Fig. The SARS-CoV-2 'testing bus'.**
(DOCX)

**S2 Fig. Distribution of the combined score of two questions on intention to vaccinate against SARS-CoV-2 among all study participants (n = 168).**
(DOCX)

## Acknowledgments

The authors would like to acknowledge all participants and employees of the homelessness service of De Regenboog Groep, Leger des Heils and HVO Querido for their close collaboration.

## Author Contributions

**Conceptualization:** Ellen Generaal, D. K. (Daniela) van Santen, Marjolein J. Booij, Dylan Price, Marcel Buster, Christa van Dijk, Yvonne T. H. P. van Duijnhoven, Maria Prins.

**Data curation:** Ellen Generaal.

**Formal analysis:** Sophie L. Campman, Anders Boyd.

**Funding acquisition:** Ellen Generaal, Mariken van der Lubben.

**Investigation:** Ellen Generaal, Marjolein J. Booij, Dylan Price, Christa van Dijk, Maria Prins.

**Methodology:** Sophie L. Campman, Maria Prins.

**Project administration:** Ellen Generaal, Marjolein J. Booij.

**Resources:** Ellen Generaal, Dylan Price, Yvonne T. H. P. van Duijnhoven.

**Supervision:** Ellen Generaal, D. K. (Daniela) van Santen, Maria Prins.

**Writing – original draft:** Ellen Generaal, Sophie L. Campman.

**Writing – review & editing:** Ellen Generaal, D. K. (Daniela) van Santen, Sophie L. Campman, Marjolein J. Booij, Dylan Price, Marcel Buster, Christa van Dijk, Anders Boyd, Sylvia M. Bruisten, Alje P. van Dam, Mariken van der Lubben, Yvonne T. H. P. van Duijnhoven, Maria Prins.

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
