## [Decision Letter · Decision Letter 0]

14 Feb 2023

PONE-D-22-35165Low prevalence of current and past SARS-CoV-2 infections among visitors and staff members of homelessness services in Amsterdam at the end of the second wave of infections in the NetherlandsPLOS ONE

Dear Dr. Generaal,

Thank you for submitting your manuscript to PLOS ONE. After careful consideration, we feel that it has merit but does not fully meet PLOS ONE’s publication criteria as it currently stands. Therefore, we invite you to submit a revised version of the manuscript that addresses the points raised during the review process.

We look forward to receiving your revised manuscript.

Kind regards,

Ian Christopher N Rocha, MD, MBA, MHSS

Academic Editor

PLOS ONE

Journal Requirements:

"The author(s) received no specific funding for this work. The research project was paid by the Public Health Service of Amsterdam." 

4. PLOS requires an ORCID iD for the corresponding author in Editorial Manager on papers submitted after December 6th, 2016. Please ensure that you have an ORCID iD and that it is validated in Editorial Manager. To do this, go to ‘Update my Information’ (in the upper left-hand corner of the main menu), and click on the Fetch/Validate link next to the ORCID field. This will take you to the ORCID site and allow you to create a new iD or authenticate a pre-existing iD in Editorial Manager. Please see the following video for instructions on linking an ORCID iD to your Editorial Manager account: https://www.youtube.com/watch?v=_xcclfuvtxQ.

Additional Editor Comments:

Dear Authors,

Thank you for submitting an interesting manuscript. Kindly address the comments and suggestions of the four reviewers.

Best regards,

Academic Editor

Reviewers' comments:

Reviewer's Responses to Questions

**Comments to the Author**

1. Is the manuscript technically sound, and do the data support the conclusions?

Reviewer #1: Yes

Reviewer #2: Yes

Reviewer #3: Yes

Reviewer #4: Yes

2. Has the statistical analysis been performed appropriately and rigorously? 

Reviewer #1: Yes

Reviewer #2: I Don't Know

Reviewer #3: Yes

Reviewer #4: No

3. Have the authors made all data underlying the findings in their manuscript fully available?

Reviewer #1: No

Reviewer #2: No

Reviewer #3: No

Reviewer #4: Yes

4. Is the manuscript presented in an intelligible fashion and written in standard English?

Reviewer #1: Yes

Reviewer #2: Yes

Reviewer #3: Yes

Reviewer #4: Yes

5. Review Comments to the Author

Reviewer #1: Very nice paper, nicely written and clearly explained. Please address the questions below to further improve clarity.

-Lines 53-55: Clarify whether antibody results were restricted to those without prior COVID-19 vaccination.

-Lines 57-58: Rephrase intention to vaccinate and trust in COVID-19 policies to indicate that the “majority” of respondents had low or medium intention to vaccinate/trust in policies.

Reviewer #2: Congratulations to the authors on the successful completion of their study.

The authors in this study analyze the prevalence of current and past SARS CoV-2 infection in People Experiencing Homelessness(PEH) in visitors and staff members of homelessness services in Amsterdam. According to the authors, the study was conducted during May 2021 when the second wave of COVID in Netherlands was over. The authors have also highlighted the fact that this is a first of its kind of study in Netherlands. The authors claim that they observed a low prevalence of past and current SARS CoV-2 infection rates among the PEH which they hypothesize can be attributed to escalated efforts on part of the dutch government to offer shelters with closed spaces for the PEH. While the manuscript presents techincally and scientifically sound data to support this claim, the inherent significance of this study could not be understood. The authors have claimed that these findings are contrary to the ones that have been previously reported from countries like France and US and even Netherlands for that matter. The authors mention that higher SARS CoC-2 prevalence in these studies could have been because of sampling from ares that had outbreaks. The findings reported are true to their core and the data presented supports it as well but a mention of the significance of conducting and publishing such a study is missing. The authors are encouraged to discuss the significance of this study in detail. And as a minor correction, the authors are advised to reframe the short title of this study.

Reviewer #3: The manuscript is well thought out and written, the objectives are clearly stated, the methods are adequate, data statistically analyzed, the conclusions well supported by the data presented. This paper is well-written and I enjoy reading through it. The authors should consider the following recommendations in order to improve the original manuscript:

[1] Abstract. Some sentences are not easy to understand: "Intention to vaccinate for COVID-19 was low (30% for visitors, 10% for staff) or medium (28% for visitors, 35% for staff). Trust in COVID-19 policies was either low (27% for visitors, 17% for staff) or neutral (32% for visitors, 53% for staff)."

[2] All participants provided written informed consent. Were there no illiterate participants? How was consent obtained from participants of Polish origin? Was there a consent form written in Polish?

[3] Since the internal consistency of the statements demonstrated high reliability (Cronbach’s α=0.95), their responses were combined as a mean score. Cronbach’s coefficient is useful to examine the internal consistency of the constructs used in the study, but they are not useful for indicating whether the mean of the responses to the items is adequate for the data analyses.

[4] Vaccination intention was categorized into low (mean score 1.0-2.5), medium (3.0-5.0) and high (5.5-7.0), based on the distribution of the combined score. Please be clearer about the criteria used to establish these cut-off points.

Reviewer #4: How does the sample size with your studies compare to other studies?

There were no formal statistical test compared

Very few patients are test positive, but does it represented the whole population?

6. PLOS authors have the option to publish the peer review history of their article (what does this mean?). If published, this will include your full peer review and any attached files.

Reviewer #1: No

Reviewer #2: No

Reviewer #3: No

Reviewer #4: No

---

## [Author Response · Author response to Decision Letter 0]

17 Apr 2023

Response to reviewers 

5. Review Comments to the Author

Reviewer #1: Very nice paper, nicely written and clearly explained. Please address the questions below to further improve clarity.

-Lines 53-55: Clarify whether antibody results were restricted to those without prior COVID-19 vaccination.

Reply: Reply: We thank the reviewer for their compliments. Antibody results were indeed restricted to those without prior COVID-19 vaccination. We added this to the abstract accordingly in line 54; “among those without prior COVID-19 vaccination”.

-Lines 57-58: Rephrase intention to vaccinate and trust in COVID-19 policies to indicate that the “majority” of respondents had low or medium intention to vaccinate/trust in policies.

Reply: We agree with the reviewer that this sentence could be improved. Since there is no large majority of respondents reporting a certain score (low or high), we rephrased this paragraph in the abstract (page 2, lines 46-48) as follows:

““High vaccination intent was more common among staff members (55%) than among visitors (42%), while high trust in the governmental COVID-19 policies was more common among visitors (41%) than among staff (30%)”

Reviewer #2: Congratulations to the authors on the successful completion of their study.

The authors in this study analyze the prevalence of current and past SARS CoV-2 infection in People Experiencing Homelessness(PEH) in visitors and staff members of homelessness services in Amsterdam. According to the authors, the study was conducted during May 2021 when the second wave of COVID in Netherlands was over. The authors have also highlighted the fact that this is a first of its kind of study in Netherlands. The authors claim that they observed a low prevalence of past and current SARS CoV-2 infection rates among the PEH which they hypothesize can be attributed to escalated efforts on part of the dutch government to offer shelters with closed spaces for the PEH. While the manuscript presents technically and scientifically sound data to support this claim, the inherent significance of this study could not be understood. The authors have claimed that these findings are contrary to the ones that have been previously reported from countries like France and US and even Netherlands for that matter. The authors mention that higher SARS CoV-2 prevalence in these studies could have been because of sampling from areas that had outbreaks. The findings reported are true to their core and the data presented supports it as well but a mention of the significance of conducting and publishing such a study is missing. The authors are encouraged to discuss the significance of this study in detail. And as a minor correction, the authors are advised to reframe the short title of this study.

Reply: We thank the reviewer for their compliments. We highlighted to the relevance of our study in the previous manuscript at the end of the Discussion, also mentioning that PEH may be at risk for more severe COVID-19. We rephrased this paragraph to update our core message, also adding a recent paper (31). 

See page 16, lines 339-344: 

“Therefore, as COVID-19 continues to circulate around the world, emergency services for PEH should be made rapidly accessible when needed, and vaccination uptake should be evaluated, in particular because PEH are regarded to be at increased risk of severe COVID-19 due to comorbidities (7, 12), including chronic infections such as hepatitis C (30), and because we found vaccination intention among PEH to be relatively low. There is growing evidence that, in addition to general preventive measures, public policies should offer adequate shelter to PEH to prevent outbreaks of infectious diseases (31).”

We agree with the reviewer that we could have mentioned the significance of our study in more detail in the introduction, and therefore added a paragraph (see page 3-4, lines 86-93 and see below for added text):

“As PCR testing of PEH may have occurred less frequently than among the general population, it is relevant to evaluate the prevalence of past SARS-CoV-2 infections in this group and among staff members working in the homelessness services who are frequently in close contact with PEH, in particular because COVID-19 continues to spread around the world and new SARS-CoV-2 variants of concern may evolve. Given that PEH may be at increased risk for severe COVID-19 due to chronic physical and mental conditions impaired by substance or alcohol abuse (7, 12, 13) and that SARS-CoV-2 transmission may occur rapidly in places with overcrowding (5, 6), studying the prevalence of past SARS-CoV-2 infections in homelessness services during the COVID-19 pandemic is highly relevant.”

Reviewer #3: The manuscript is well thought out and written, the objectives are clearly stated, the methods are adequate, data statistically analyzed, the conclusions well supported by the data presented. This paper is well-written and I enjoy reading through it. The authors should consider the following recommendations in order to improve the original manuscript:

Reply: We thank the reviewer for their compliments; your review and input is highly appreciated. 

[1] Abstract. Some sentences are not easy to understand: "Intention to vaccinate for COVID-19 was low (30% for visitors, 10% for staff) or medium (28% for visitors, 35% for staff). Trust in COVID-19 policies was either low (27% for visitors, 17% for staff) or neutral (32% for visitors, 53% for staff)."

Reply: We agree with the reviewer and changed this sentence in the abstract (see page 2, lines 46-48 and previous point of reviewer 1):

“High vaccination intent was more common among staff members (55%) than among visitors (42%), while high trust in the governmental COVID-19 policies was more common among visitors (41%) than staff (30%)”

[2] All participants provided written informed consent. Were there no illiterate participants? How was consent obtained from participants of Polish origin? Was there a consent form written in Polish?

Reply: Thank you for this interesting question. Indeed, we made sure that all reading material, including the respondents information and informed consent forms, was translated into Polish (and English). A Polish-to-English translator was present at both research sites to make sure that the study information was understood by the respondent. No respondents were excluded due to illiteracy, although we noticed that some PEH went over the forms very quickly and did not read very attentively. Therefore, we additionally used posters with pictures of the different study elements (PCR swab, blood fingerstick, questionnaire), that we pointed out to make sure that all the research steps and the risk of having to go into isolation came across as best as possible. Our interpretation was that no PEH felt having to go into isolation (a single room for one week) as a major barrier, let alone a motivating factor to participate. Staff members of one homelessness service assisted with pre-selecting PEH for the research study, hence PEH that seemed cognitively unable to participate or were under the strong influence of alcohol/drugs, were not invited by the staff. This may have led to no actual exclusions at study entry.

We added a few sentences to clarify this in the manuscript:

Methods (page 4-5, lines 114-116): At one testing location, including the majority of participants, staff members pre-selected visitors who met inclusion criteria…. “For this reason, and because we had an onsite translator, we did not have to exclude any potential participant.”

Methods (page 5, lines 119-120): “Research information and consent forms were translated into English and Polish, and a Polish-to-English translator was present during the study.”

[3] Since the internal consistency of the statements demonstrated high reliability (Cronbach’s α=0.95), their responses were combined as a mean score. Cronbach’s coefficient is useful to examine the internal consistency of the constructs used in the study, but they are not useful for indicating whether the mean of the responses to the items is adequate for the data analyses.

Reply: Thank you for this remark. We did not primarily base the combination of both scores on Cronbach’s alpha but this was rather an theoretical rationale. We changed this sentence accordingly:

Methods (page 6, lines 145-150): “We combined the responses of the two statements as a mean score to create a more accurate measure of vaccination intent, based on Ajzen’s Theory of Planned Behavior which states that both behavioral intentions and perceived behavioral control can predict behavior (14). Additionally, the internal consistency of the two statements demonstrated high reliability (Cronbach’s α=0.95).”

[4] Vaccination intention was categorized into low (mean score 1.0-2.5), medium (3.0-5.0) and high (5.5-7.0), based on the distribution of the combined score. Please be clearer about the criteria used to establish these cut-off points.

Reply: The cut-off points were selected based on the distribution of the variable as indicated in the supplementary figure, S4 Figure. This was also in line with the method used in a previous study (ref. 16).

To clarify this, we added one reference (see page 6, lines 149-150): “Vaccination intention was categorized into low (mean score 1.0-2.5), medium (3.0-5.0) and high (5.5-7.0), based on the distribution of the combined score (S5 Figure) and in line with another study(16).”

To clarify our cut-off points; both statements could only be scored as integers, thus when combining these scores as a mean, this would only result in either an integer, such as 2.0, or half a score, such as 2.5 (a mean score between 2.0 and 2.5 is not possible). 

Reviewer #4: How does the sample size with your studies compare to other studies?

There were no formal statistical test compared

Very few patients are test positive, but does it represented the whole population?

Reply: Thank you for these questions. 

We had a lower sample size in our study as compared to other studies, but given that the PEH community is small in the Netherlands, we still reached 3% (n=138) of the estimated total homeless population in Amsterdam (approximately n=5000 PEH, personal communication (ref. 21)). As a comparison, in Marseille France (Ly et al, Int J Infect Dis. 2021), the number of PEH was estimated at 14,063 in 2016 and the authors screened 698 PEH in their study in 2020, resulting in a rate of 5%. However, this estimated rate is likely lower when taking into account that the number of PEH after 2016 increased in Marseille, resulting in more similar proportions of the target population. 

Referring to the second question, we indeed did not formally test to compare the proportion of current and past SARS-CoV-2 infections between staff members and visitors. This was not the aim of our study, which was more descriptive of nature, taking into account the relatively small sample size.

We added a comment on the generalizability of our study to the Discussion (page 15, lines 330-333):

“Whether our results are generalizable to other PEH in the Netherlands cannot be guaranteed, however we did make an effort to reduce selection bias by including both walk-in centers (more frequently visited by people living on the streets) and temporary housing for PEH, aiming to include a diverse group of PEH in our study.”

We thank the reviewers for their adequate feedback.

Kind regards,

Ellen Generaal

---

## [Decision Letter · Decision Letter 1]

2 Jul 2023

Low prevalence of current and past SARS-CoV-2 infections among visitors and staff members of homelessness services in Amsterdam at the end of the second wave of infections in the Netherlands

PONE-D-22-35165R1

Dear Dr. Generaal,

We’re pleased to inform you that your manuscript has been judged scientifically suitable for publication and will be formally accepted for publication once it meets all outstanding technical requirements.

Kind regards,

Ian Christopher N Rocha, MD, MBA, MHSS

Academic Editor

PLOS ONE

Additional Editor Comments (optional):

Reviewers' comments:

Reviewer's Responses to Questions

**Comments to the Author**

1. If the authors have adequately addressed your comments raised in a previous round of review and you feel that this manuscript is now acceptable for publication, you may indicate that here to bypass the “Comments to the Author” section, enter your conflict of interest statement in the “Confidential to Editor” section, and submit your "Accept" recommendation.

Reviewer #1: All comments have been addressed

2. Is the manuscript technically sound, and do the data support the conclusions?

Reviewer #1: Yes

3. Has the statistical analysis been performed appropriately and rigorously? 

Reviewer #1: Yes

4. Have the authors made all data underlying the findings in their manuscript fully available?

Reviewer #1: No

5. Is the manuscript presented in an intelligible fashion and written in standard English?

Reviewer #1: Yes

6. Review Comments to the Author

Reviewer #1: All comments have been adequately addressed. Reasons for not making data available need to be made explicit in the Data Availability Statement.

7. PLOS authors have the option to publish the peer review history of their article (what does this mean?). If published, this will include your full peer review and any attached files.

Reviewer #1: No

---

## [Editor Report · Acceptance letter]

14 Jul 2023

PONE-D-22-35165R1 

Low prevalence of current and past SARS-CoV-2 infections among visitors and staff members of homelessness services in Amsterdam at the end of the second wave of infections in the Netherlands 

Dear Dr. Generaal:

I'm pleased to inform you that your manuscript has been deemed suitable for publication in PLOS ONE. Congratulations! Your manuscript is now with our production department. 

Kind regards, 

on behalf of

Dr. Ian Christopher N Rocha 

Academic Editor

PLOS ONE